# Relative Importance Analysis of Safety Climate Evaluation Factors Using Analytical Hierarchical Process (AHP)

**Hyunjin Lim, Sunkuk Kim** 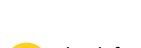**, Yonggu Kim and Seunghyun Son ***

Department of Architectural Engineering, Kyung Hee University, Yongin-si 17104, Korea;
agency0023@khu.ac.kr (H.L.); kimskuk@khu.ac.kr (S.K.); ygkim09@khu.ac.kr (Y.K.)
* Correspondence: seunghyun@khu.ac.kr; Tel.: +82-31-201-3685

**Abstract:** Various studies have confirmed that the increasing quality of safety climate has a positive influence on reducing the occurrence of accidents. The quality of safety climate is comprehensively affected in three domains: management, site, and enterprise. At the company level, it is challenging to manage all areas at a high level due to limited managerial resources. Therefore, it is necessary to establish a strategy that improves the safety climate step by step. For the efficient execution of the strategy, it is necessary to analyze the relative importance of each evaluation factor of the safety climate and allocate managerial resources accordingly. Therefore, this study aims to analyze the relative importance of safety climate evaluation factors using the analytical hierarchical process (AHP) technique. For this study, AHP questionnaire and analysis are conducted, and the relative priorities of safety climate evaluation factors are derived. As a result, (E) workers' safety priority and risk non-acceptance is the most important dimension among seven dimensions as the weight is 0.1900. In addition, (E1) compliance with safety regulations, even if the process is tight, is the most important one between items as the weight 0.6663. The results of this study will be used as basic data for institutional improvement and policy making for a high-quality safety climate at construction sites.

**Keywords:** safety climate; relative importance analysis; analytical hierarchical process (AHP); safety management



## 1. Introduction

The construction industry still has a relatively high injury and death rate compared to other industries [1–4], despite many efforts to ensure worker safety [5,6]. According to statistics from the Korea Occupational Safety and Health Agency, the construction industry mortality rate surged 2.13‰ to 9.41‰ in 2018 from 7.28‰ in 2014 and has been continuously increasing every year [7]. In particular, in 2019, out of 855 worker fatalities in all industries, the number of worker fatalities in the construction industry was 428, accounting for more than 50% of the total [8]. This shows that safety management in construction sites is not sufficient. In recent construction projects, a new approach is taken in terms of reducing accidents, which is creating and establishing a safety climate among workers [9–14]. Various studies have confirmed that the increasing quality of the safety climate has a positive influence on reducing the occurrence of accidents [15–19]. Zohar [15] stated that when a safety climate is established in a construction site, workers will behave carefully to comply with safety guidelines and prevent accidents. Bronkhorst [16] found that when a high-quality safety climate is formed in a workplace, it has a positive effect on the safety behavior of workers, and the accident frequency is reduced. If we look at the fundamental cause of safety accidents in terms of the level of safety awareness of the members of an organization, efforts must be made to improve the safety climate of construction sites for sustainable disaster prevention in the construction industry. According to the NOSACQ-50 questionnaire developed by the National Research Center for Work Environment (NR-CWE) [20], the quality of safety climate is comprehensively affected in three domains:

management, site, and enterprise. At the company level, it is challenging to manage all areas at a high level due to limited managerial resources. Therefore, it is necessary to establish a strategy that improves the safety climate step by step. To improve the safety climate, it is necessary for a company to analyze the relative importance of each evaluation factor of the safety climate and allocate managerial resources accordingly. For example, if workers' safety priority and risk nonacceptance are found to be the most important factors among the top seven dimensions for evaluating safety climate, the management shall prioritize increasing the level of workers' safety priority and risk nonacceptance within available resources. At this time, the company shall establish a strategy to ensure compliance with safety regulations even if the process is tight, which is of high relative importance among the two items of workers' safety priority and risk nonacceptance. Analyzing the relative importance of safety climate evaluation factors in this way can suggest a standard for resource allocation for efficient safety management. However, measuring the quality of the safety climate cannot simply be evaluated quantitatively based on the physical and economic points of view [21,22]. Since subjective opinions are reflected in the evaluation of safety climate, it would be difficult to maintain consistency unless the evaluation is conducted by an experienced evaluator such as a safety expert [23,24]. For this reason, a safety expert who has a direct influence on workers' safety awareness in the field and who is in actual charge of safety management should evaluate the safety climate [25]. Therefore, this study aims to analyze the relative importance of safety climate evaluation factors using the analytical hierarchical process (AHP) technique. To this end, the study proceeds as shown in Figure 1.

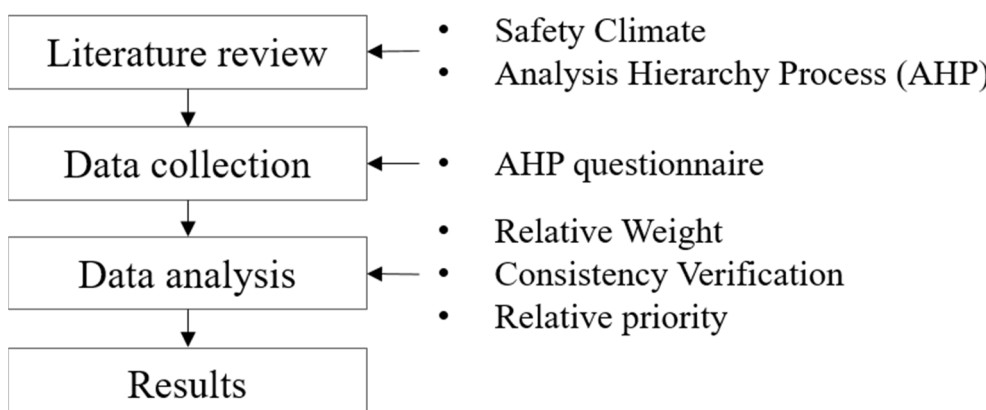

**Figure 1.** Methodology.

First, preliminary studies are conducted related to the safety climate of the construction industry and the AHP technique. Second, an AHP questionnaire drive is conducted for safety experts at construction sites in Korea. Third, an AHP analysis is performed using the collected data to analyze the relative importance of the safety climate evaluation factors. Based on this, the relative priorities of factors are derived. Fourth, using the analysis results, an improvement plan is proposed for the improved safety climate of construction sites in Korea. Through this, an improvement plan is proposed for a better safety climate of construction sites in Korea. Currently, the research related to the safety climate simply measured the level of the safety climate [26,27] or analyzed the correlation between productivity and the factors [28]. In other words, although the safety climate is affected to the construction safety, it is difficult to know the relative importance between factors for improving safety climate in real site. Therefore, this study analyzed the relative importance of the factors. The findings can suggest criteria to establish the resource input strategy for improving safety climate in real site. The results of this study will be used as basic data for institutional improvement and policy making for high-quality safety climates at construction sites.

## 2. Preliminary Study

### 2.1. Concept of Safety Climate

Zohar [29] suggested the safety climate concept firstly, and various research was conducted regarding the characteristics of the safety climate [9–19,30,31]. Safety climate is the comprehensive perception of workers regarding the value of safety such as policy, process, and custom within the organization. In addition, the safety climate is a sub-facet of the organizational culture influencing the workers' behavior related to safety in the organization. In particular, according to Mohmad [32], safety climate is a set of shared perceptions of the work environment of managers and workers, representing workers' perception of the value of safety in work environments. Therefore, safety climate has a great influence on behaviors and attitudes related to the safety of individuals or an organization and can be used as a decisive tool for predicting the possibility of workplace accidents [33–35]. When examining literature related to safety climate, safety climate established within an organization is closely related to the occurrence of workplace accidents because it affects various work factors related to safety. Moreover, it has a significant influence on workers' awareness and behavior about safety [30–33,36,37]. In particular, in the case of construction sites where workers' unsafe behavior is considered the biggest cause of accidents [38], it is important to improve workers' safety behavior and attitudes by creating a safety climate within the organization [39–41] to reduce accidents. Fang, Chen, and Wong [42] analyzed the correlation between safety climate and safety behavior in construction sites in Hong Kong. The results showed that the more firmly established safety climate is, the more likely it is for workers to comply with safety guidelines. Furthermore, the level of safety awareness was also much higher for sites where the safety climate is better established than those where it is not. This means that safety climate in the construction industry has a great influence on workers' awareness and attitudes about safety, and it is necessary to create a safety climate in workplaces to reduce workplace accidents in the construction industry. In the research applying to construction sites regarding the safety climate, He et al. [35] analyzed the correlation between the safety climate such as management commitment, safety knowledge and coworker perception, and safety behavior such as safety compliance and safety participation. As a result, the safety climate is affected by safety behavior. Lingard, Pirzadeh, and Oswald [41] analyzed the correlation between the social network metrics of subcontracted construction workgroups and the safety climate. As a result, communication skills play a pivotal role to improve the safety climate. In addition, the study shows that the social network between management and workers has to be established. In this respect, the importance of the safety climate is increased between workers for reducing the accident rate in the current construction projects, thus, this study analyzes the relative importance of evaluation factors to effectively improve the safety climate. The findings can support effective safety management under the limited managerial resources in a real site.

### 2.2. Evaluation of Safety Climate Using AHP Analysis

Industrial accidents occur frequently due to workers' carelessness caused by insufficient and unreliable evaluation systems of safety culture and risk factors [43,44]. In addition, it is difficult to maintain the consistency of evaluation because the subjective opinions of workers are reflected in the process in the existing evaluation method. There is also a limitation in decreased quality of safety culture due to the extended cycle of safety evaluations [45]. In this regard, the AHP analysis technique is needed to develop quantitative evaluation indicators through rational and consistent decision making [46,47]. The AHP technique is a technique that classifies a number of properties hierarchically, identifies the importance of each property, and then selects the optimal alternative based on the result [48,49]. Through the technique, the weights of elements that are initially difficult to quantify can be derived step by step, and difficult information can be processed relatively easily [50]. The evaluation results based on the AHP technique are highly reliable in terms of consistency and integrity and can be used as a rational decision-making tool in that it analyzes the relative importance and priorities between elements by stratification and sim-

plification [51,52]. The process of AHP analysis is as shown in Figure 2. First, the hierarchy of decision making is established, and the objective of the most comprehensive decision making in the top layer is placed; then, the next layers are consisted to the factors affecting the objective [47,49]. Second, the weights are calculated by pairwise comparison with nine-point scale [48,50]. Third, the consistency of the survey result is reviewed. If it has no consistency, the feedback is needed. Fourth, the weight of an alternative is determined.

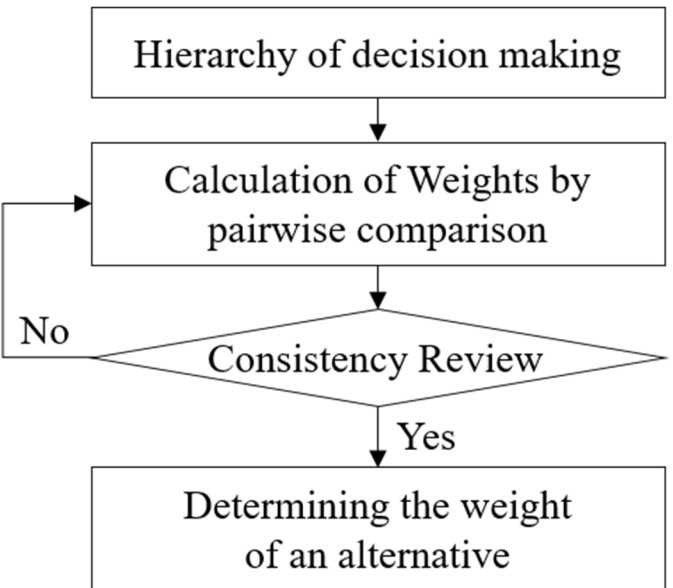

**Figure 2.** Procedure of analytical hierarchical process (AHP) application.

In particular, in order for safety climate and safety culture evaluation tools to be used for the safety management system in a workplace, it is necessary to objectively judge the relative importance of each item and to derive quantitative values accordingly. Through this, the relative weights of evaluation items for a safety climate can be calculated, and their effectiveness could be verified by reviewing the results [46,53]. Therefore, this study aims to evaluate the safety climate and safety culture of construction sites by using the AHP technique in order to increase the reliability of safety climate evaluation, and to minimize errors stemming from subjective judgment. Then, based on the evaluation results, an improvement plan will be proposed for better safety climate.

### 3. Data Collection

*3.1. AHP Analysis Overview*

This study conducted a survey of safety experts working at 48 construction sites in Korea. The survey period was for 6 months from March to September in 2019. Of the 48 questionnaires surveyed, only 25 with a consistency index of 0.1 or less were selected. The demographics of the sample are shown in Table 1. Sampling was conducted with 25 safety managers. Among them, 6 were in their 30s (24%), 6 were in their 40s (24%), and 13 (52%) were in their 50s or older. There was 1 person who had less than 5 years of experience (4%), 3 people who had 5–10 years of experience (12%), and 21 people who had more than 10 years of experience (84%). As shown in Table 1, 84% of the samples were surveyed by a group of experts who had been in charge of safety management for more than 10 years. An AHP analysis was performed using samples with a consistency index within 0.1. That is, in this study, a consistent sample of survey results was used for a group of experts in safety management. Therefore, the results derived from this study are considered highly reliable.

**Table 1.** Demographics.

| Variable | Category | N | % |
|---|---|---|---|
| Age | 30–39 years | 6 | 24 |
| | 40–49 years | 6 | 24 |
| | ≥50 years | 13 | 52 |
| Career | ≤5 years | 1 | 4 |
| | 6–9 years | 3 | 12 |
| | ≥10 years | 21 | 84 |

*3.2. Analysis Factors*

In AHP analysis, the evaluation criteria and alternatives are organized in a hierarchical structure, and then, the priority is determined by deriving relative importance [54,55]. In this study, in order to analyze the relative importance of the safety climate evaluation factors, the evaluation factors should be stratified. The decision-making issues of this study can be stratified as shown in Figure 3.

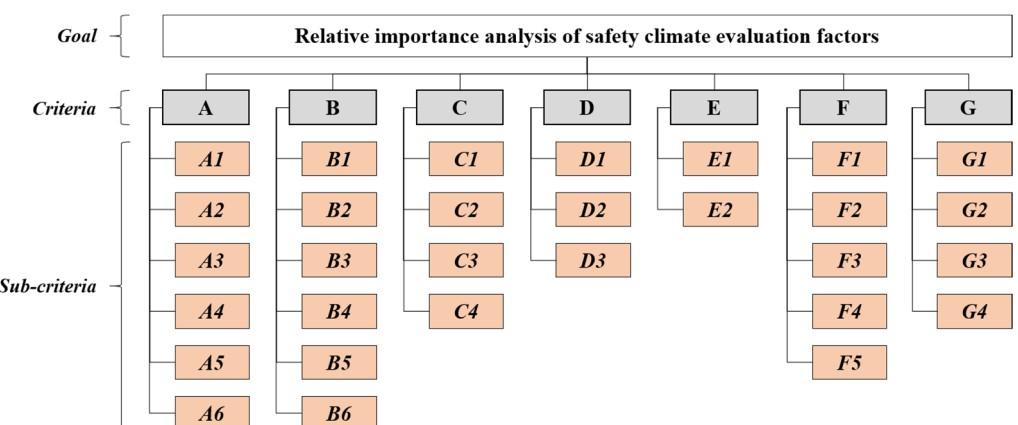

**Figure 3.** Hierarchical structure.

As shown in Figure 3, the goal of AHP analysis in this study is to derive the relative importance of safety climate evaluation factors. This is composed of 7 dimensions and 30 items. Refer to Table 2 for details on evaluation items for each level in Figure 3. In this study, after analyzing the relative importance of dimensions of the first level, the items of each dimension are analyzed. Finally, the relative priorities of the evaluation factors are derived, and improvement measures are proposed for an improved safety climate.

The safety climate evaluation factors in Figure 3 were set up based on the NOSACQ-50 questionnaire, which is most commonly used in measuring safety climate. NOSACQ-50 is a questionnaire developed to measure the quality of safety climate of an organization in Nordic countries, and it consists of a total of 50 questions on 7 safety climate assessment dimensions [56]. The NOSACQ-50 questionnaire consists of positively formulated items and reversed formulated items for one item in order to review whether the respondent responded consistently [57]. In this study, based on NOSACQ-50, 7 dimensions were composed as shown in Table 2. A total of 30 items were selected among 50 evaluation items excluding overlapping questions due to reversed formulated items through brainstorming with safety experts. Table 2 summarizes the safety climate evaluation factors for AHP analysis.

**Table 2.** Detailed factors for evaluating safety climate [56,57].

| Dimension | Questionnaire |
|---|---|
| A. Management safety priority and commitment, and competence | A1. Compliance with management's safety policy<br>A2. Provision of all safety information<br>A3. Safety management system maintenance<br>A4. Safety considerations rather than productivity<br>A5. Whether field managers trust the management's safety management capabilities<br>A6. Management's actions in detecting risks at the site |
| B. Management safety empowerment | B1. Efforts by the management on regular safety inspections<br>B2. Workers participate in decisions that affect their safety<br>B3. Consideration of workers' suggestions for safety<br>B4. Training of site managers on safety<br>B5. Collect opinions from site managers<br>B6. Field managers' participation in safety decisions |
| C. Management safety justice | C1. Efforts to collect information in the event of an accident<br>C2. Efforts to listen to field managers' opinions in the event of an accident<br>C3. Efforts to actively investigate the cause of an accident in the event of an accident<br>C4. Investigate the accident fairly with the field manager in the event of an accident |
| D. Workers' safety commitment | D1. Joint responsibility for site safety of field participants<br>D2. Interest in the safety of individual field participants<br>D3. Mutual efforts of field participants to ensure safe working |
| E. Workers' safety priority and risk nonacceptance | E1. Compliance with safety regulations even if the process is tight<br>E2. If fieldwork violates the safety regulations, report to the upper part |
| F. Safety communication, learning, and trust in co-workers' safety competence | F1. Trying to find a solution to a safety issue when it is pointed out<br>F2. Mutual trust in safety during collaboration<br>F3. Training from previous accident experiences among field participants to prevent accidents<br>F4. Interchange opinions of field participants and reflect them in the work<br>F5. Active discussion among field participants on safety |
| G. Trust in the general efficacy of safety systems | G1. Consideration to ensure that safety systems play a major role in preventing accidents<br>G2. Continuous implementation of regular safety education<br>G3. Implementation of a safety plan<br>G4. Establish clear objectives for safety systems |

NOSACQ-50, as the evaluation tool of safety climate, is classified into seven dimensions as shown in Table 2. Among them, (A) management safety priority and commitment, and competence, (B) management safety empowerment, (C) management safety justice have to be evaluated in management domain. In addition, (D) workers' safety commitment, (E) workers' safety priority and risk non-acceptance, (F) safety communication, learning, and trust in co-workers' safety competence have to be evaluated in site domain. Especially, (G) trust in the general efficacy of safety systems has to be evaluated in enterprise domain including managements and workers together. Figure 4 describes for readers to understand easily regarding the evaluation factors of seven dimension in NOSACQ-50.

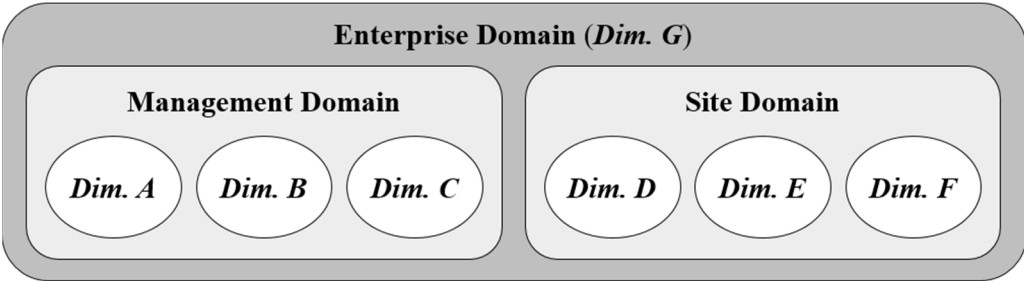

**Figure 4.** Classification of factors for evaluating safety climate.

Lee et al. [28] analyzed the correlation between safety climate level and construction productivity quantitatively. In this study, the NOSACQ-50 survey was conducted to measure the safety climate level at site, and the level was evaluated in management, site, and enterprise domain. However, this study proved only the correlation between safety climate level and construction productivity and the strategy to improve safety climate was not suggested. Son et al. [26] analyzed the safety climate level of the construction company in each business size. This study was also conducted the NOSACQ-50 survey to evaluate the safety climate level. As a result, the level was analyzed differentially according to organizational safety culture. This result represented that safety climate is not evaluated only site domain, but it has to be considered in management and enterprise domain together. He et al. [27] analyzed the safety climate of construction workers in South Korea by using NOSACQ-50. As a result, in the case of South Korea, the managements consider productivity more than safety. In addition, workers do not trust the judgment and responsibility of managements.

In this way, many studies measured the safety climate level of construction site by using NOSACQ-50. However, detailed strategies were not suggested to improve the safety climate level. Especially, it is difficult that all domains are maintained to a high level at the same time because of limited managerial resources. Therefore, this study identifies the relative importance of the evaluation factors of NOSACQ-50, and the strategy establishment can be used for improving the safety climate.

In this study, a pairwise comparisons matrix as shown in Table 3 was used to analyze the relative importance of the evaluation items in Table 2. The evaluation of the items was conducted using the 9-point scale proposed by Satty [58] as shown in Table 4. As shown in Table 3, the comparison matrix of this study takes the form of an inverse centered around the diagonal. When the values of the two factors in the matrix are 1, it means that they have equal relative importance [58]. In addition, the size of the comparison matrix may change depending on the number of factors in the hierarchy [50].

**Table 3.** Matrix of paired comparisons.

|  | $F_1$ | $F_2$ | $F_3$ | $F_4$ | $\cdots$ | $F_n$ |
|---|---|---|---|---|---|---|
| $F_1$ | 1 | $F_1/F_2$ | $F_1/F_3$ | $F_1/F_4$ | $\cdots$ | $F_1/F_n$ |
| $F_2$ | $F_2/F_1$ | 1 | $F_2/F_3$ | $F_2/F_4$ | $\cdots$ | $F_2/F_n$ |
| $F_3$ | $F_3/F_1$ | $F_3/F_2$ | 1 | $F_3/F_4$ | $\cdots$ | $F_3/F_n$ |
| $\vdots$ | $\vdots$ | $\vdots$ | $\vdots$ | 1 | $\vdots$ | $\vdots$ |
| $\vdots$ | $\vdots$ | $\vdots$ | $\vdots$ | $\vdots$ | $\vdots$ | $\vdots$ |
| $F_n$ | $F_n/F_1$ | $F_n/F_2$ | $F_n/F_3$ | $F_n/F_4$ | $\cdots$ | 1 |

**Table 4.** AHP scale for combinations.

| Scale | Definition | Explanation |
|---|---|---|
| 1 | Equally important | Both factors have the same criteria. |
| 3 | Moderately important | One criterion is slightly more important than the other. |
| 5 | Strongly important | One criterion is more important than the other. |
| 7 | Very strongly important | One criterion is far more important than the other. |
| 9 | Extremely important | One criterion is extremely more important than the other. |
| 2, 4, 6, 8 | The median of the two scales | The median of the two scales. |

As shown in Table 4, the 9-point scale proposed by Satty [32] can relatively easily process not only quantitative information such as the years of experience or intuition of the questioners but also qualitative information that must be taken into account [59].

The safety climate of a site, which is the subject of this study, represents the level of safety awareness of workers, and it is very subjective. Therefore, the information should be handled qualitatively. Therefore, in this study, in order to maintain the consistency of the results, an AHP questionnaire was administered to a group of safety experts who have performed safety management tasks for a long period of time. The relative importance of the safety climate evaluation factors was analyzed based on the results.

## 4. Data Analysis

Table 5 shows the relative importance and priority measurement results for the dimension of the safety climate evaluation factor at case sites. (E) workers' safety priority and risk nonacceptance, as the weight is 0.1900, is the highest priority between the measurement indicators, then the priority shows (D) workers' safety commitment (0.1887), (F) safety communication, learning, and trust in co-workers' safety competence (0.1847), (A) management safety priority and commitment, and competence (0.1634), (G) trust in the general efficacy of safety systems (0.1303), (B) management safety empowerment (0.0740), and (C) management safety justice (0.0688), sequentially.

**Table 5.** Relative importance and priority of the dimensions in safety climate.

| Code | Dimension | Weight | Priorities |
|---|---|---|---|
| E | Workers' safety priority and risk nonacceptance | 0.1900 | 1 |
| D | Workers' safety commitment | 0.1887 | 2 |
| F | Safety communication, learning, and trust in co-workers' safety competence | 0.1847 | 3 |
| A | Management safety priority and commitment, and competence | 0.1634 | 4 |
| G | Trust in the general efficacy of safety systems | 0.1303 | 5 |
| B | Management safety empowerment | 0.0740 | 6 |
| C | Management safety justice | 0.0688 | 7 |

In the case sites surveyed as shown in Table 5, it was found that workers' safety priority and risk nonacceptance (E) was relatively the most important. The detailed items for item E are "compliance with safety regulations even if the process is tight" and "if fieldwork violates the safety regulations, report to the upper part." In a construction project, the most ideal form is to proceed with a project based on the initial plan without any change until the completion of the project. However, in actual construction projects, schedule changes or cost increases are inevitable because site conditions change due to the request of the client or faulty work. In order to create and spread the safety climate at the site, the safety manager should comply with safety regulations even in such conditions and

should thoroughly report to upper-level managers about any behavior that violates the safety regulations.

In addition, Dimension D, which evaluates the level of joint responsibility and mutual effort of site workers, and Dimension F, which evaluates whether workers communicate about safety and their mutual trust in safety, were evaluated as top priorities. Dimension A, which evaluates whether managers comply with safety policies and prioritizes safety over productivity, and Dimension G, which evaluates whether the safety system plays a major role in preventing accidents, were evaluated as mid-range priorities. Dimension B, which evaluates whether management considers site manager's suggestions for safety and reflects them properly when making a decision, and Dimension C, which evaluates whether the cause of an accident is actively investigated and the opinions of site managers are heard in the event of an accident, was rated as low range priorities.

Additionally, the local and global relative importance and priority measurements on items in safety climate factors are shown in Table 6. First of all, looking at the relative importance and priority of local sites, in the case of the items of E, D, and F, which were evaluated as top priorities, the items such as "compliance with safety regulations even if the process is tight (0.6663)", "mutual efforts of field participants to ensure safe working (0.5119)", and "interchange opinions of field participants and reflect them in the work (0.2699)" were found to be relatively important. In the items of A and G, which were evaluated as mid-range priorities, the items "management's actions in detecting risks at the site (0.2044)" and "consideration to ensure that safety systems play a major role in preventing accidents (0.3359)" turned out to be relatively important. In addition, in the case of the items of B and C that were evaluated as low-range priorities, the items "field managers' participation in safety decisions (0.2164)" and "efforts to actively investigate the cause of an accident in the event of an accident (0.3217)" were found to be relatively important.

**Table 6.** Relative importance and priority of the items of safety climate.

| Dimension | Item | Local Weight | Priorities | Global Weight | Priorities |
|---|---|---|---|---|---|
| A. Management safety priority and commitment, and competence (0.1634) | A1. Compliance with management's safety policy | 0.1723 | 3 | 0.0282 | 14 |
| | A2. Provision of all safety information | 0.1278 | 6 | 0.0209 | 20 |
| | A3. Safety management system maintenance | 0.1485 | 5 | 0.0243 | 18 |
| | A4. Safety considerations rather than productivity | 0.1868 | 2 | 0.0305 | 13 |
| | A5. Whether field managers trust the management's safety management capabilities | 0.1601 | 4 | 0.0262 | 16 |
| | A6. Management's actions in detecting risks at the site | 0.2044 | 1 | 0.0334 | 12 |
| B. Management safety empowerment (0.0740) | B1. Efforts by the management on regular safety inspections | 0.1388 | 6 | 0.0103 | 29 |
| | B2. Workers participate in decisions that affect their safety | 0.1994 | 2 | 0.0147 | 24 |
| | B3. Consideration of workers' suggestions for safety | 0.1395 | 5 | 0.0103 | 29 |
| | B4. Training of site managers on safety | 0.1487 | 4 | 0.0110 | 28 |
| | B5. Collect opinions from site managers | 0.1571 | 3 | 0.0116 | 27 |
| | B6. Field managers' participation in safety decisions | 0.2164 | 1 | 0.0160 | 23 |

**Table 6.** *Cont.*

| Dimension | Item | Local Weight | Priorities | Global Weight | Priorities |
|---|---|---|---|---|---|
| C. Management safety justice (0.0688) | C1. Efforts to collect information in the event of an accident | 0.2007 | 3 | 0.0138 | 25 |
| | C2. Efforts to listen to field managers' opinions in the event of an accident | 0.1934 | 4 | 0.0133 | 26 |
| | C3. Efforts to actively investigate the cause of an accident in the event of an accident | 0.3217 | 1 | 0.0222 | 19 |
| | C4. Investigate the accident fairly with the field manager in the event of an accident | 0.2842 | 2 | 0.0196 | 21 |
| D. Workers' safety commitment (0.1887) | D1. Joint responsibility for site safety of field participants | 0.2435 | 3 | 0.0459 | 6 |
| | D2. Interest in the safety of individual field participants | 0.2447 | 2 | 0.0462 | 5 |
| | D3. Mutual efforts of field participants to ensure safe working | 0.5119 | 1 | 0.0966 | 2 |
| E. Workers' safety priority and risk nonacceptance (0.1900) | E1. Compliance with safety regulations even if the process is tight | 0.6663 | 1 | 0.1266 | 1 |
| | E2. If fieldwork violates the safety regulations, report to the upper part | 0.3337 | 2 | 0.0634 | 3 |
| F. Safety communication, learning, and trust in co-workers' safety competence (0.1847) | F1. Trying to find a solution to a safety issue when it is pointed out | 0.2060 | 3 | 0.0381 | 9 |
| | F2. Mutual trust in safety during collaboration | 0.1905 | 4 | 0.0352 | 10 |
| | F3. Training from previous accident experiences among field participants to prevent accidents | 0.1043 | 5 | 0.0193 | 22 |
| | F4. Interchange opinions of field participants and reflect them in the work | 0.2699 | 1 | 0.0499 | 4 |
| | F5. Active discussion among field participants on safety | 0.2293 | 2 | 0.0423 | 8 |
| G. Trust in the general efficacy of safety systems (0.1303) | G1. Consideration to ensure that safety systems play a major role in preventing accidents | 0.3359 | 1 | 0.0438 | 7 |
| | G2. Continuous implementation of regular safety education | 0.1920 | 4 | 0.0250 | 17 |
| | G3. Implementation of a safety plan | 0.2641 | 2 | 0.0344 | 11 |
| | G4. Establish clear objectives for safety systems | 0.2081 | 3 | 0.0271 | 15 |

Examining the global relative importance and priorities that comprehensively consider all the items, in order to improve the safety atmosphere at construction sites, items such as "compliance with safety regulations even if the process is tight (0.1266)", "mutual efforts of field participants to ensure safe working (0.0966)", "if fieldwork violates the safety regulations, report to the upper part (0.0634)", "interchange opinions of field participants and reflect them in the work (0.0499)", and "interest in the safety of individual field participants (0.0462)" must be improved by management and field manager's cooperative efforts.

## 5. Discussion

In this study, an AHP questionnaire was conducted for 25 safety managers to analyze the relative importance of safety climate evaluation factors. As a result, (E) workers' safety priority and risk nonacceptance, as the weight is 0.1900, is the highest one among dimensions and (C) management safety justice, as the weight is 0.0688, is the lowest one. (E1) compliance with safety regulations even if the process is tight, as the weight is 0.1266, shows the highest one in the relative importance considering all items. (B1) efforts by the

management on regular safety inspections and (B3) consideration of workers' suggestions for safety are the lowest one, as the weight is 0.0103. In this respect, the managements can perform safety management effectively under limited resources if the budget is arranged based on the suggested relative priority in each items.

Management will be able to efficiently perform safety management under limited resources if they reflect the derived relative priorities of each factor when budgeting for safety management. For example, Lee, Son, Kim, and Son [28] quantitatively analyzed the correlation between safety climate and construction productivity. They argued that productivity increases as the level of safety climate increases. However, site managers believe that it is difficult to manage all areas within limited resources, given the circumstances where additional costs are inevitable to raise the level of safety. If the relative importance of the safety climate evaluation factors proposed by this study is utilized, the management would be able to allocate resources according to priorities. Son et al. [26] and Ha et al. [27] analyzed the level of workers' awareness of safety climate by project and by size. Using the relative importance of the safety climate evaluation factors in this study additionally, further studies could be conducted on the effect of reduced accidents and improved productivity resulting from additional safety management costs and the enhanced quality of safety climate at a site. The results of this study can present the criteria for resource allocation for efficient safety management and can support management's decision making on safety.

In the future, the research could be conducted regarding the accident rate reduction and productivity improvement according to the additional safety management cost and the improvement of safety climate level by using relative importance of the evaluation factors as the findings from this study. Therefore, the findings of this study can suggest the criteria of resource distribution for effective safety management and support the decision making related to safety management of managements. However, only 25 projects were analyzed due to limited time and budget in this study. In the future, the reliability could improve through collecting the data from more projects. The results of this study will serve as basic data for the development of a simulation model that can easily and quickly predict increased productivity according to the improvement of the quality of climate safety.

## 6. Conclusions

This study aims to analyze the relative importance of safety climate evaluation factors using the AHP technique. To do this, the AHP questionnaire drive and analysis were conducted for safety experts of construction sites, and based on this, relative priorities of safety climate evaluation factors were derived. Through this, criteria for efficient operation were proposed based on the relative priority of each management item so that companies could improve safety climate under limited managerial resources. The results of this study are as follows:

First, the relative importance of the dimensions in the safety climate evaluation was analyzed. The analysis result showed that "workers' safety priority and risk nonacceptance (0.1900)" was the highest followed by "workers' safety commitment (0.1887)", "safety communication, learning, and trust in co-workers' safety competence (0.1847)", "management safety priority and commitment, and competence (0.1634)", "trust in the general efficacy of safety systems (0.1303)", "management safety empowerment (0.0740)", and "management safety justice (0.0688)." The derived results provide safety management priorities for the dimensions, and management can use them to establish measures to prevent accidents.

Second, the relative importance of the items in the safety climate evaluation was analyzed. As a result, the case of the items of E, D, and F, which were evaluated as top priorities, the items such as "compliance with safety regulations even if the process is tight (0.6663)", "mutual efforts of field participants to ensure safe working (0.5119)", and "interchange opinions of field participants and reflect them in the work (0.2699)" were found to be relatively important. In the items of A and G, the items "management's actions in detecting risks at the site (0.2044)" and "consideration to ensure that safety systems

play a major role in preventing accidents (0.3359)" turned out to be relatively important. In addition, in the case of the items of B and C, the items "field managers" participation in safety decisions (0.2164)" and "efforts to actively investigate the cause of an accident in the event of an accident (0.3217)" were found to be relatively important. The derived results provide safety management priorities for the items, and management can use them to establish measures to prevent accidents.

Third, global relative importance and priority were analyzed by considering all items comprehensively. As a result, in order to improve safety atmosphere at construction sites, compliance with safety regulations even if the process is tight (0.1266)", "mutual efforts of field participants to ensure safe working (0.0966)", "if fieldwork violates the safety regulations, report to the upper part (0.0634)", "interchange opinions of field participants and reflect them in the work (0.0499)", and "interest in the safety of individual field participants (0.0462)" were found to be relatively important. Using the derived results, management can allocate resources according to priorities when calculating safety management costs to improve the safety climate.

As such, this study presents criteria for resource allocation for efficient safety management within the limited resources of the company. Using the results derived, additional research, such as reduction in safety accidents and productivity improvement studies, can be conducted according to the improvement of the safety climate level. In the future, the results of this study will be used as a basis for institutional improvement and policy establishment for a high-quality safety climate at a construction site.

**Author Contributions:** Conceptualization, H.L. and S.S.; methodology, H.L. and S.S.; validation, S.K. and Y.K.; formal analysis, S.S. and Y.K.; investigation, H.L.; resources, H.L.; data curation, H.L.; writing—original draft preparation, H.L., S.S., Y.K., and S.K.; writing—review and editing, S.S. and S.K.; visualization, S.S. and S.K.; supervision, S.K. All authors have read and agreed to the published version of the manuscript.

**Funding:** This work was supported by the National Research Foundation of Korea (NRF) grant funded by the Korea government (MOE) (No. 2017R1D1A1B04033761).

**Institutional Review Board Statement:** Not applicable.

**Informed Consent Statement:** Informed consent was obtained from all subjects involved in the study.

**Data Availability Statement:** Data sharing is not applicable to this article.

**Conflicts of Interest:** The authors declare no conflict of interest.

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
