# Peer review of "Relative Importance Analysis of Safety Climate Evaluation Factors Using Analytical Hierarchical Process (AHP)"

_sustainability, doi:10.3390/su13084212_

Round 1
Reviewer 1 Report
This study examined importance of safety climate evaluation factors using Analytical Hierarchical Process (AHP).
- Abstract
The abstract should include purpose of this study, method, results, and implications in belief. However, the abstract highlights the research topic, problem statement, and the purpose of this study. Please check the abstract and revise to summarize the manuscript.
- Citation
I would like to suggest that the authors need to use appropriate citations in this manuscript. I think some sentences miss citations.
- Evaluation of safety climate
The definition of constructs (i.e., management, site, and enterprise) in the Safety climate measurement would be helpful to understand the measurement. The examples of previous research on using the measurement are also necessary.
Moreover, ‘2.2. evaluation of safety climate’ focuses on the AHP analysis. The title can be revised.
- Figure 3
Why is other dimensions nested in Dimension G?
Is there any problems if all constructs are analyzed in the same level?
Does the authors propose the Figure 3?
If the source is provided, it would be helpful for understanding the concepts.
- Data analysis
Instead of using ‘>’, the results may need to account for Table 5 in different way.
- Discussion
Discussion should include more rich information about the findings of this study.
- Limitations and future suggestions
Limitations and future suggestions can be guided to the future researchers who are interested in this study.
Author Response
"Please see the attachment"

Reviewer 2 Report
Ref: sustainability-1155717
Journal: Sustainability (ISSN 2071-1050), MDPI
Title: Relative Importance Analysis of Safety Climate Evaluation Factors using Analytical Hierarchical Process (AHP)
- General comments
- This study is interesting and aims to analyze the relative importance of safety climate evaluation factors using the AHP technique.
- The topic of this manuscript is appropriate for Sustainability (ISSN 2071-1050), MDPI journal
- The overall presentation of the paper is good
- The paper’s length is suitable for contribution
- I find that the paper is a useful contribution to the original research
- The article contains an extensive analysis of source materials, a description of the methodology and the whole manuscript illustrates an interesting study.
- The literature list includes over 35 items.
- Nevertheless, this is not to say that it is without blemish. I have several comments regarding the scientific part of the manuscript and I expect the authors to proceed in a resubmission after an efficient reformation of their article.
- In the current version of the manuscript, the authors did not explain clearly what the novelty and key points of this study are.
- A nomenclature incorporated in the end of the manuscript, would be very useful.
- Moreover, taking into consideration that there are several limitations in this study due to the assumptions made, a suggestion for further research, and also some proposed scientific directions, have to be incorporated.
- The manuscript’s similarity-index (by using the TURNITIN checker) is 29%, and after excluding bibliography is 18%.
- Specific comments
Abstract:
- The abstract could be reformed in order to present in a clearer way, the deductions of the work.
- The main aim of the paper is ambiguous, so the abstract has to be reformed in order to present in a clear way, the purpose of the work.
- Introduction:
- There are some paragraphs which are identical with the ones coming from the literature and could be replaced or rephrased, as for example the following: Lines 79-87
- The introduction motivates the study, quite all right, but if possible please make it clearer to the reader, i.e. what is the new scientific contribution of your work.
- In the presented literature review, the authors did not explain what the limitations are of previous research and what the authors are going to improve that.
- The literature review is incomplete, and it should be completed by more recent studies.
- A more thorough presentation of the literature review about AHP is essential.
Discussion:
- This section is quite cursory, and could be revised. The authors did not discuss clearly on what the insights are from this new study and how these insights can be generalized to other situations.
References
- The literature review is incomplete, and it should be completed by more recent studies.
Decision: My opinion is the article must be resubmitted, after revisions of the manuscript.
Attached document: Please see the attached documents with reviewer’s comments.

Reviewer 3 Report
The article addresses and interesting topic that fits the scope of Sustainability. The paper is well put together and the applied methodology is interesting. However, some important issues must be addressed in order to improve the overall quality of the work: 1) Figure 1 quality must be improved, as it is poor. 2) Address the implications of a small sample for the overall research results (section 3.1). 3) Figure 2 quality can be improved. 4) Figure 3 quality must be improved, as it is poor. 5) Table 5 must be stretched in order to allow the last column title not to be split between two lines. 6) The discussion and conclusion sections are rather small and should be expanded. This is particularly important for the discussion section (only 14 lines; 1 paragraph). 7) The greatest limitation of the paper: the references and what they say about the thoroughness with which the author performed the literature review: a. In 36 references, only 6 are recent (5 years or less); it is strongly suggested that the author must improve this ratio. b. Most of the references are regional in nature; there are a great deal of recent, non-regional (global) literature that is being ignored by the authors; the text would improve greatly if enriched with some additional sources with those characteristics.Author Response
Please see the attachment

Round 2
Reviewer 1 Report
Thank you for your revision.
Reviewer 2 Report
Ref: sustainability-1155717
Journal: Sustainability (ISSN 2071-1050), MDPI
Title: Relative Importance Analysis of Safety Climate Evaluation Factors using Analytical Hierarchical Process (AHP)
The authors would like to sincerely appreciate the anonymous reviewer who provided thorough reviews and valuable comments to help us improve the manuscript. We strongly believe that in the revision we have fully addressed the reviewer’s comments and concerns and carefully revised the manuscript based on the feedback we have received. Please see the followings below responding to the reviewer’s comments.
General comments
This study is interesting and aims to analyze the relative importance of safety climate evaluation factors using the AHP technique.
The topic of this manuscript is appropriate for Sustainability (ISSN 2071-1050), MDPI journal
The overall presentation of the paper is good
The paper’s length is suitable for contribution
I find that the paper is a useful contribution to the original research
The article contains an extensive analysis of source materials, a description of the methodology and the whole manuscript illustrates an interesting study.
The literature list includes over 35 items.
Nevertheless, this is not to say that it is without blemish. I have several comments regarding the scientific part of the manuscript and I expect the authors to proceed in a resubmission after an efficient reformation of their article.
In the current version of the manuscript, the authors did not explain clearly what the novelty and key points of this study are.
A nomenclature incorporated in the end of the manuscript, would be very useful.
Moreover, taking into consideration that there are several limitations in this study due to the assumptions made, a suggestion for further research, and also some proposed scientific directions, have to be incorporated.
The manuscript’s similarity-index (by using the TURNITIN checker) is 29%, and after excluding bibliography is 18%.
Specific comments
Abstract:
The abstract could be reformed in order to present in a clearer way, the deductions of the work.
The main aim of the paper is ambiguous, so the abstract has to be reformed in order to present in a clear way, the purpose of the work.
[Response] We revised the abstract as follows by reflecting reviewer's comments.
Ok. The authors revised the manuscript (comparatively with the previous submission), as follows:
Revised. Line 11~24.
Various studies have confirmed that the increasing quality of safety climate has a positive influence on reducing the occurrence of accidents. The quality of safety climate is comprehensively affected in three domains: management, site, and enterprise. At the company level, it is challenging to manage all areas at a high level due to limited managerial resources. Therefore, it is necessary to establish a strategy that improves the safety climate step by step. For the efficient execution of the strategy, it is necessary to analyze the relative importance of each evaluation factor of the safety climate and allocate managerial resources accordingly. Therefore, this study aims to analyze the relative importance of safety climate evaluation factors using the AHP technique. For this study, AHP questionnaire and analysis are conducted and the relative priorities of safety climate evaluation factors are derived. As a result, (E) workers’ safety priority and risk non-acceptance is the most important dimension among seven dimensions as the weight is 0.1900. In addition, (E1) compliance with safety regulations even if the process is tight is the most important one between items as the weight 0.6663. The results of this study will be used as basic data for institutional improvement and policy-making for a high-quality safety climate at construction sites.
Introduction:
There are some paragraphs which are identical with the ones coming from the literature and could be replaced or rephrased, as for example the following: Lines 79-87
[Response] We revised the paragraphs as follows by reflecting the reviewer's comments.
Ok. The authors revised the manuscript (comparatively with the previous submission), as follows:
Revised. Line 83~87.
The introduction motivates the study, quite all right, but if possible please make it clearer to the reader, i.e. what is the new scientific contribution of your work.
In the presented literature review, the authors did not explain what the limitations are of previous research and what the authors are going to improve that.
[Response] We added the following to the paper by reflecting the reviewer's comments.
Ok. The authors revised the manuscript (comparatively with the previous submission).
The literature review is incomplete, and it should be completed by more recent studies.
A more thorough presentation of the literature review about AHP is essential.
[Response] We added the following to the paper by reflecting the reviewer's comments.
Ok. The authors revised the manuscript (comparatively with the previous submission).
Discussion:
This section is quite cursory, and could be revised. The authors did not discuss clearly on what the insights are from this new study and how these insights can be generalized to other situations.
[Response] We added the following to the paper by reflecting the reviewer's comments.
Ok. The authors revised the manuscript (comparatively with the previous submission), as follows:
Revised. Line 279~310.
In this study, an AHP questionnaire was conducted for 25 safety managers to analyze the relative importance of safety climate evaluation factors. As a result, (E) workers’ safety priority and risk non acceptance as the weight is 0.1900 is the highest one among dimensions and (C) Management safety justice as the weight is 0.0688 is the lowest one. (E1) compliance with safety regulations even if the process is tight as the weight 0.1266 shows the highest one in the relative importance considering all items. (B1) efforts by the management on regular safety inspections and (B3) consideration of workers' suggestions for safety are lowest one as the weight 0.0103. In this respect, the managements can perform safety management effectively under limited resources if the budget is arranged based on the suggested relative priority in each items.
Management will be able to efficiently perform safety management under limited resources if they reflect the derived relative priorities of each factor when budgeting for safety management. For example, Lee, Son, Kim, and Son [34] quantitatively analyzed the correlation between safety climate and construction productivity. They argued that productivity increases as the level of safety climate increases. However, site managers believe that it is difficult to manage all areas within limited resources, given the circumstances where additional costs are inevitable to raise the level of safety. If the relative importance of the safety climate evaluation factors proposed by this study is utilized, the management would be able to allocate resources according to priorities. Son et al. [35] and Ha et al. [36] analyzed the level of workers’ awareness of safety climate by project and by size. Using the relative importance of the safety climate evaluation factors in this study additionally, further studies could be conducted on the effect of reduced accidents and improved productivity resulting from additional safety management costs and the enhanced quality of safety climate at a site. The results of this study can present the criteria for resource allocation for efficient safety management and can support management’s decision-making on safety.
In the future, the research can conduct regarding the accident rate reduction and productivity 302 improvement according to the additional safety management cost and the improvement of safety 303 climate level by using relative importance of the evaluation factors as the findings from this study. 304 Therefore, the findings of this study can suggest the criteria of resource distribution for effective safety 305 management and support the decision making related to safety management of managements. 306 However, only 25 projects were analyzed due to limited time and budget in this study. In the future, 307 the reliability can improve through collecting the data from more projects. The results of this study will 308 serve as basic data for the development of a simulation model that can easily and quickly predict 309 increased productivity according to the improvement of the quality of climate safety.
References
The literature review is incomplete, and it should be completed by more recent studies.
Ok. The authors revised the manuscript (comparatively with the previous submission).
Decision: My opinion is the article could be accepted for publication in its current version.
Attached document: Please see the attached document including the reviewer’s comments.

Reviewer 3 Report
Dear authors,
I consider you have made and met most of my improvement suggestions. This has greatly improved the quality and clarity of the text. Therefore, I now consider that the content succinctly described and contextualized with respect to previous and present theoretical background and empirical research, that the research design, questions, hypotheses and methods reasonably stated, that the arguments and discussion of findings is minimal coherent and balanced, that the empirical research and the results are adequately presented and that the conclusions are, to some degree, supported by the results presented in the article.
The article has, now, conditions to be published.
Best regards.